# Research on Customer Behavioral Intention of Hot Spring Resorts Based on SOR Model: The Multiple Mediation Effects of Service Climate and Employee Engagement

Zhengyan Guo *  , Yao Yao and Yuan-Cheng Chang

Chinese International College, Dhurakij Pundit University, Bangkok 10210, Thailand;
629890020008@dpu.ac.th (Y.Y.); yuan-cheg.cha@dpu.ac.th (Y.-C.C.)
* Correspondence: guo52tvxq@163.com; Tel.: +86-185-6729-0092

**Abstract:** Based on the stimulus–organism–response (SOR) model, this study explored customer behavioral intentions and influencing factors in the service industry, represented by hotels. It studied the servicescape, customer emotions, and customer behavioral intentions. PROCESS analysis was conducted on 305 valid questionnaires collected from hot spring resorts. The study found that servicescape can predict customer behavioral intentions, that customer emotions have a partial mediating effect in the influence of servicescape on customer behavioral intentions, and that service climate and employee engagement have multiple mediation effects in the SOR model. Therefore, the researchers suggest that hotels can improve customer perceptions through decoration and staff management, and thereby develop the hotel in a sustainable way.

**Keywords:** SOR model; customer behavioral intentions; servicescape; customer emotion; multiple mediation effects; sustainable hospitality management

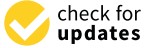



## 1. Introduction

### 1.1. Introduction

Service is an important element for any company that aims to increase its popularity and attract customers; for example, the well-known Chinese hot pot brand Haidilao is famous for its service. The development of hotels depends on good service to an even greater extent. The service industry has surpassed manufacturing in bringing intangible benefits to the economy, and building customer loyalty in the marketplace has become increasingly important and challenging [1]. As key players in the service industry, hotels are also facing this major challenge, and establishing customer loyalty has become the focus of many hotel managers. To address this issue, researchers in environmental psychology are increasingly focusing on how the environment affects the service experience of new and existing customers, and market researchers are using the physical environment of hotels as an important evaluation indicator [2]. To achieve the sustainable development of hotels in today's international environment, it is important to increase the focus on the impact of our services on customers. This study provides suggestions for the sustainable development of a hotel by studying the factors influencing customer behavior at a hot spring resort.

There are many factors that influence customer behavioral intentions. This study considers the influence of hotels on customer behavioral intentions in terms of both service environment and customer emotions. A service environment is a collective and shared space that is based on organizational practices focused on customer service [3]. Additionally, a service environment demonstrates a company's or firm's commitment to customers and service [4]. Thus, a service environment embodies an organization's readiness to provide excellent service. In this study, servicescape, service climate, and employee engagement are used to represent the service environment of the hotel. In his study, Bitner [5] introduced the servicescape, a concept that helped subsequent researchers understand

the important role of the environment, and, more specifically, the physical environment. Wakefield and Blodgett [6] conducted a study on the important role of the leisure service environment, and the results clearly showed that this is important not only to customers but also to the company. Reimer and Kuehn [7] found that it was essential to include the physical environment in the quality assessment of leisure services similar to hotels. This environmental psychology concept of a company's physical environment is referred to as the servicescape—a variable that describes the customer's overall experience of the entire service and facility. Servicescape is not only an important component in the formation of customer impressions, but also an important source of evidence for the overall assessment of the service industry and the organization [8]. Service climate is an indicator of whether a firm has sufficient skills and resources to provide high quality services to customers [9]. In business operations, the effectiveness of a company's service climate is an important factor in its ability to coordinate the management of the company [9,10]. If the service climate is operating at a high level, employees have the energy to solve the problems encountered by customers. Similarly, when the quality of service is encouraged by the organization, employees will put effort into overcoming the challenges posed by customers [9]. In contrast, if the service climate is operating at a lower level, hotels lack the resources and skills required to provide quality service to customers [11]. Some researchers have argued that employee behavior is important to the customer experience [12–14]. Researchers have also suggested that service climate and employee engagement are the most discussed topics in organizations that promote high quality customer experience and service [15]. Because employees play such an important role in the service industry, the knock-on effects of their attitudes and the environmental conditions can also have an impact on customers.

Therefore, the purpose of this study was to apply an integrated architecture—the SOR Model—to a research case using hotels to represent the service industry. In response to consumers' increasing environmental consciousness, and to lessen the sector's negative effects on the environment, society, and the economy, green hotels and sustainable practices have received much research. We would like to use this research to provide managers with management ideas in terms of servicescape, customer emotion, and employee engagement to help hotels achieve green hotels and sustainable development.

### 1.2. Theoretical Basis

The theoretical basis for this study is the stimulus–organism–response (SOR) model proposed by Mehrabian and Russell [16], which is the most widely used model in the environmental psychology literature and consumer intention studies [17]. In this model, stimuli (S) from the environment affect an individual's internal assessment (O) [18], which in turn affects their behavior (R) [16]. The SOR model is mainly used to explain the effect of external environmental stimuli on individuals' cognitive and emotional states, and thus on behavioral responses [16]. The model proposes that people learn by relying on sensation and perception, and by drawing on the subjective organization in the brain to function mechanistically, rather than through trial and error. In contrast with the traditional stimulus–response model, the S-O-R model posits that the connection between stimulus and response is not direct and mechanical, but that there is an intermediary link in between, and that the entire behavioral process is governed by perception. Individual perception is a necessary process for responding to environmental stimuli [19].

The full SOR model contains stimulus, organism perception, and response variables. In recent years, the SOR model has been used more frequently in the study of consumer behavior. Donovan and Rossiter [20] first applied the SOR model to the context of consumer behavior, and they studied the influence on customer behavior using the retail environment as a stimulus variable and the customer's perceived situation as a mediator. In this study, in the decision-making process of consumer behavioral tendencies, the stimulus originated from the product characteristics–service landscape; customer sentiment represented the consumer emotion and perception factors; and the consumer's behavioral outcome was finally produced after having been influenced, i.e., the customer behavioral intention [21].

Liang, Li, and Sun [22] studied the impact of organizational learning on organizational innovation performance based on the SOR model. In their study of online product displays, Zheng, Hu, and Han [23] used functional features of these displays, among others, to represent external factor stimuli. Shi, Meng, and Li [24] used the SOR model to study consumers' purchasing intentions in online purchasing and found that the online environment and the risk perceived by the consumers had a significant influence on their purchasing decisions. Liao, Wong, Palvia, and Kakhki [25] explained the role of the SOR model and found that an online store display was one of the key determinants and emotional stimuli that triggered the desire of consumers to make impulse purchases. Sultan, Jan, Basit, and Rafiq [26] defined impulse buying behavior by implementing an SOR model in which store ambience was a key determinant of consumer impulse spending through positive emotions. Many researchers have studied consumer stimuli, emotional responses, and impulse purchases by adapting the SOR model [27–30]. All these studies show that the SOR model is convincing for the study of consumer behavior. Our study used perceived servicescape as an influencing factor and customer emotion as a psychological perception and measured their role in influencing the behavioral intentions of customers.

## 2. Literature Review

Customers are now extremely sensitive to new service evaluation methods. The hospitality industry is very competitive, making it difficult for hotel marketers to comprehend consumer behavior [31]. In order to better suggest a comprehensive and sustainable development for the hotel industry, we propose corresponding hypotheses based on the literature.

Servicescape is an important component of the service industry. It can establish immediate perceptual impressions in the minds of customers and can therefore have a significant impact on them [8,32]. The servicescape encompasses the impression of the overall product and service of the organization. Because the hospitality industry provides highly intangible products such as services, consumers are likely to judge and assess a hotel based on tangible aspects such as its appearance [33]. In this case, the servicescape is not only an integral part of the customer's impression, but is an important source of evidence enabling the customer to assess the organization as a whole. In the SOR model, individual behavior is generated by emotional stimulation and modulation caused by stimuli in the environment, which cause the body to produce either acceptance or rejection. The impact of the servicescape on the customer stems first and foremost from his or her own visual perceptions. The customer's visual perception of the quality of maintenance and core service values of a hotel represents the actual service level of the hotel [34]. The degree of arousal can be as adequate a representation of one's emotional state, such as pleasure/displeasure [35]. In his study of hot spring resorts, Chang [36] found that customers were more demanding regarding the hotel amenities, and that the servicescape of the physical environment could be pleasurable for customers. Babin and Attaway [37] found that consumers' positive perception of the physical environment of the service landscape could trigger positive emotions and lasting satisfaction. Peng, Wang, and Lam [38] investigated the servicescape of a tourist-integrated resort in China and found that the servicescape and customers' behavioral intentions were positively correlated, and recommended that resort managers should focus on the design of the servicescape. Kampani and Jhamb [39] found that the servicescape of a beauty salon could influence customers' behavioral intentions and moderate this behavior through customers' emotional states. Liao [40] also found that customers influence behavioral intentions not only through their attitudes, but also through the perceived landscape. Thbillejas-Andres, Cervera-Taulet, and García [41], in a study of similar service industry venues, found that the service landscape directly and indirectly influenced the audience's post-consumption, with emotions playing a partially mediating role. Positive emotions promote customers' willingness to consume. Wang and Kim [42] found that customers' behavioral intentions are influenced by customers' emotions, and that they will only spend if the services provided by

the merchants make them feel positive emotions. Cha and Shin [43] also found that, among other factors, an increase in customers' emotional response makes them more willing to buy. Based on previous studies and the S-O-R model, we propose the following hypotheses:

**H1.** *Servicescape has a positive impact on customer emotions.*

**H2.** *Customer emotions have a positive impact on customer behavioral intentions.*

**H3.** *Servicescape has a positive impact on customer behavioral intentions.*

Based on the SOR model, service climate and employee engagement were added to this study. Service climate refers to the employees' perception of the service environment and practices [44]. Service climate is an important indicator of employee motivation and satisfaction and customer behavior [11]. Conceptually, the difference between service climate and servicescape is that service climate emphasizes human perception, whereas servicescape is more concerned with the physical environment. The servicescape affects the feelings of both employees and customers, and the information employees receive about the importance of service in the organization can affect their performance [45]. A good working environment enables employees to experience their value to the organization or enterprise, which will produce a positive service atmosphere.

In the hot spring resort environment, in addition to the physical environment that affects customers' emotions, the service climate is another major experience for customers [36]. In their study of service marketing, He, Li, and Keung Lai [46] argued that organizations must create and maintain an atmosphere that encourages employees to effectively provide quality services, and that this should be the theme of service marketing. In the process of service consumption, employees and customers inevitably come into contact, and therefore human contact forms the basis of the service climate influence. It has been inferred that a possible mechanism by which service climate influences customer perception is the actual interpersonal behavior of employees when interacting with customers [47]. Increasing interpersonal and emotional interaction requires more frequent contact activities. The service environment is fundamental to the well-being of employees. Employees will put more effort into providing a quality service to satisfy customers if they are happy, and, to be happy, they require access to specific resources from their service environment that meet their job needs [48]. Therefore, the following hypotheses are proposed:

**H4.** *Servicescape has a positive impact on the service climate.*

**H5.** *Service climate has a positive impact on customer emotions.*

In addition, employees and customers are frequently in contact during service consumption. For this reason, Dietz et al. [47] argued that human contact is the basis for influencing service climate and inferred that the possible mechanism by which service climate affects customer perceptions is the actual interpersonal behavior of employees when interacting with customers, and that to experience more interpersonal and emotional interaction requires more frequent contact activities. Accordingly, the service environment of the employees' workplace is a precursor to work well-being, and employees who invest a lot of energy in providing a quality service to satisfy customers will be happy if they have access to specific resources from their service environment that meet their work needs [48]. As Bakker and Demerouti [49] concluded, dedicated employees are energetic and passionate about their work. An energetic organizational climate is the result of positive employee engagement, in which information is shared freely and openly, relationships between employees working together are stable, and employees support and encourage each other, so that the organization benefits from dynamic employee relationships [50]. Thus, employee engagement is the extent to which employees are committed to and engaged with their organization and its values. Employees have measurable positive or negative emotional attachments to their jobs, colleagues, and organizations, which greatly influence their willingness to learn and perform at work [51]. Since employee engagement has a considerable impact on customer satisfaction with a company, the only way to maintain consistently

high levels of customer loyalty is to maintain a workforce that is so enthusiastic, creative, and energetic that the company outperforms its competitors in terms of service [52]. To date, there is no single, universally accepted definition of employee engagement, which Gallup [53] defines as engagement with and enthusiasm for work, that is closely related to organizational performance outcomes. Specifically, companies with energetic, motivated employees go further to create a superior experience for their customers, who in turn reward the company with high levels of loyalty and contribute to their profit growth. Dickson [54] suggests that fostering employee engagement in the hospitality industry is worthwhile because the industry is characterized by low salaries and predominantly short-term positions. Ncube and Jerie [55] conducted a study in the hospitality industry to understand how best to use employee engagement as a source of human resource planning and competitive advantage. Markos and Sridevi [53] found that employee engagement was related to employee passion for and commitment to the company, as well as their willingness to invest in themselves, and that employee engagement could help employers and companies succeed. Swarnalatha and Prasanna [56] revealed that engaged employees were willing to dedicate themselves to accomplishing tasks that were important for the achievement of organizational goals. Zeithaml [13] et al. also found that engaged employees would work harder to deliver on the service promises that the company offered to its customers and that customer expectations would therefore be more successfully met. In this study, employee engagement was based on the customer's perception of employee engagement, as it was assessed using surveys of customers.

In summary, although the customer's perception of the servicescape affects their emotions during service consumption, the emotions of the employees will influence the service experience. The service climate and employee engagement will be perceived by customers as part of the environment of the premises and will also have an indirect effect on their behavioral intentions. Consequently, our study made the following hypotheses:

**H6.** *Customer emotions have a positive impact on employee engagement.*

**H7.** *Employee engagement has a positive impact on customer behavioral intentions.*

Figure 1 illustrates the research framework of this study.

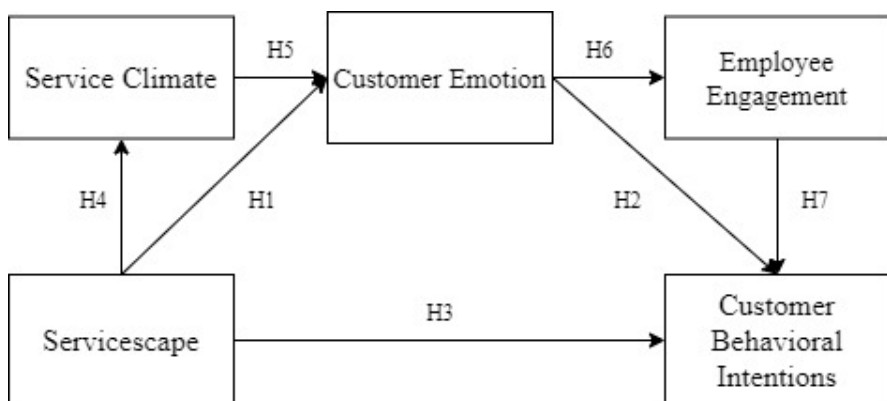

**Figure 1.** Research Framework.

## 3. Methods

### 3.1. Participants

In this study, a convenience sampling method was used to conduct the survey. The research objects were customers of a hot spring resort in Taiwan. The questionnaire was completed anonymously, and the research was known and agreed to by the participants. A total of 314 questionnaires were distributed. After discarding invalid questionnaires, 305 questionnaires remained, of which 49.8% were completed by males and 50.2% by females, giving a male to female ratio close to 1:1. The majority of subjects were aged 25–34 (51.5%), followed by 18–24 (25.5%), and 35–44 (20.8%). Of the subjects, 49.8% visited

2–4 times per year, 20.3% visited 5–6 times per year, 19.5% visited up to once a year, and another 10.4% visited more than 7 times per year.

### 3.2. Instruments

Servicescape (SS). We used the 8-item questionnaire used by Chang [12] in his study to measure servicescape. The questions were sourced from the research questionnaire created by Dong and Siu [57] to measure servicescape, using a 7-point Likert scale from 1 (strongly disagree) to 7 (strongly agree) with a Cronbach's $\alpha$ of 0.92. The Cronbach's $\alpha$ for the servicescape in this study was 0.911.

Customer Emotions (CE). We used four items proposed by Lin and Liang [58] in their study to specifically measure customer emotions. These four items were drawn from Hennig-Thurau, Groth, Paul, and Gremler [59], and were elated, peppy, enthusiastic, and excited, with a Cronbach's $\alpha$ of 0.85. In this study, the Cronbach's $\alpha$ for the customer emotions scale was 0.918.

Customer Behavioral Intentions (BI). The three items from Kuo, Chang, Chen, and Hsu's [60] study were used to measure customer behavioral intentions on a 7-point Likert scale ranging from 1 (strongly disagree) to 7 (strongly agree). The Cronbach's $\alpha$ for customer behavioral intention in this study was 0.817.

Service Climate (SC). A simplified version of the Global Service Climate Scale was used, with four questions and a Cronbach's alpha of 0.84 on a 7-point Likert scale from 1 (strongly disagree) to 7 (strongly agree) [11]. The Cronbach's $\alpha$ for service climate in this study was 0.894.

Employee Engagement (EE). Four items that were designed to measure employee engagement in the study by Britt, Castro, and Adler [61] were used on a 7-point Likert scale from 1 (strongly disagree) to 7 (strongly agree). In this study, the Cronbach's $\alpha$ for employee engagement was 0.864.

The demographic variables at the end of the questionnaire included gender, age, and number of visits, with a Cronbach's $\alpha$ of 0.970 for the overall questionnaire.

Prior to statistical analysis, researchers conducted a confirmatory factor analysis (CFA) for the questionnaire. The CFA results showed that the overall model of this questionnaire had $\chi^2$ = 402.419, $p$ = 0.000, $\chi^2/df$ = 2.022, meeting the <3 criterion. Tanaka's study [62] showed that almost all studies were significant when the sample size was greater than 200 ($p < 0.05$); both the absolute fitness indicators, GFI = 0.893 and AGFI = 0.865, met the $\geq$0.80 criterion, indicating that the model improved well in terms of fit to the observed information; the root mean squared error of approximation (RMSEA) = 0.058, meeting the criterion of <0.08, indicating that the model had a good fit; all the value-added fitness indicators met the criteria, NFI = 0.932, IFI = 0.964, CFI = 0.964, and RFI = 0.921, indicating that the model had a good degree of improvement in fitness compared to the standalone model; finally, the simple calibration fitness metrics were PNFI = 0.803 and PGFI = 0.703, both of which met the criteria, indicating that the model was acceptable.

With regard to the reliability and validity of the questionnaire (Table 1), the CR of each potential variable was greater than 0.6, AVE was greater than 0.5, and Cronbach's $\alpha$ for each variable was greater than 0.70. There was a moderate correlation between each of the two variables. These indicated that the scale had good internal consistency and stability. Next, the researchers conducted a differential validity analysis (Table 2). In Table 2, $\Delta\chi^2$ were all significant, indicating that each potential variable had differential validity among them.

**Table 1.** Reliability and validity analysis (*n* = 305).

| Variable | SS | CE | BI | SC | EE |
|---|---|---|---|---|---|
| SS | - | | | | |
| CE | 0.739 *** | - | | | |
| BI | 0.766 *** | 0.838 *** | - | | |
| SC | 0.780 *** | 0.830 *** | 0.859 *** | - | |
| EE | 0.774 *** | 0.835 *** | 0.885 *** | 0.900 *** | - |
| M | 4.959 | 4.352 | 4.650 | 4.753 | 4.712 |
| SD | 1.270 | 1.414 | 1.514 | 1.529 | 1.553 |
| Cronbach's α | 0.911 | 0.918 | 0.817 | 0.894 | 0.864 |
| CR | 0.918 | 0.821 | 0.865 | 0.896 | 0.911 |
| AVE | 0.586 | 0.606 | 0.682 | 0.685 | 0.720 |

*** $p < 0.001$. For ease of expression, in Tables 1 and 2, SS is an abbreviation for servicescape, CE is customer emotions, BI is customer behavioral intentions, SC is service climate, and EE is employee engagement.

**Table 2.** Summary table of differential validity among potential variables.

| Variable Comparison Mode | Restricted Mode ($\varphi_{ij}$ = 1) | | Standard Mode ($\varphi_{ij}$ = Free) | | $\Delta\chi^2$ |
|---|---|---|---|---|---|
| | $\chi^2$ | df | $\chi^2$ | df | |
| SC vs. EE | 495.543 *** | 200 | 402.419 *** | 199 | 93.124 *** |
| SC vs. SS | 495.543 *** | 200 | 402.419 *** | 199 | 45.575 *** |
| SC vs. AF | 447.994 *** | 200 | 402.419 *** | 199 | 44.513 *** |
| SC vs. BI | 446.932 *** | 200 | 402.419 *** | 199 | 76.605 *** |
| EE vs. SS | 479.024 *** | 200 | 402.419 *** | 199 | 28.347 *** |
| EE vs. AF | 430.766 *** | 200 | 402.419 *** | 199 | 30.925 *** |
| EE vs. BI | 433.344 *** | 200 | 402.419 *** | 199 | 58.910 *** |
| SS vs. AF | 461.329 *** | 200 | 402.419 *** | 199 | 13.596 *** |
| SS vs. BI | 416.015 *** | 200 | 402.419 *** | 199 | 27.650 *** |
| AF vs. BI | 430.069 *** | 200 | 402.419 *** | 199 | 31.893 *** |

*** $p < 0.001$.

### 3.3. Analysis Methods

The statistical analysis of this study was carried out using SPSS 23.0 software (IBM, Armonk, NY, USA). To assess the model of multiple mediation effects of the hot spring resort servicescape on customer behavioral intentions, we used Model 6 (3 mediators) of PROCESS, available in SPSS syntax by Hayes [63], for the empirical data analysis. The research hypotheses were tested by obtaining regression coefficients. Compared to the widely used Sobel Test, the PROCESS analytical model is better able to identify the significance of indirect effects [64], and its predictive validity results are fairly accurate [65]. The numerous models included in PROCESS reflect its broad applicability to meet the needs of this study. In the next step, the Bootstrap Method was used to test the 95% confidence interval and to determine whether it contained 0. If it contained 0, the indirect effect would be considered insignificant; if it did not contain 0, this would indicate the presence of a significant indirect effect. The multiple mediation effects of customer emotion, service climate, and employee engagement were tested using this procedure.

## 4. Analysis and Results

The purpose of this study was to investigate the multiple mediation effects of service climate and employee engagement using a PROCESS analysis.

### 4.1. Analysis of Multiple Mediation Effects

To examine the mediating effects of customer emotions, service climate, and employee engagement between servicescape and customer behavioral intentions, this study used Model 6 in PROCESS 3.4 to conduct a regression analysis of the mediation effects (Table 3). The results showed that: servicescape positively predicted customer emotions

(β = 0.233, *p* = 0.000); servicescape significantly positively predicted customer behavioral intentions (β = 0.766, *p* = 0.000); customer emotions positively predicted customer behavioral intentions (β = 0.246, *p* = 0.000); servicescape significantly positively predicted service climate (β = 0.780, *p* = 0.000); service climate positively predicted customer emotions (β = 0.648, *p* = 0.000); customer emotions positively predicted employee engagement (β = 0.249, *p* = 0.000); and employee engagement positively predicted customer behavioral intentions (β = 0.434, *p* = 0.000). In addition, the overall effect model $R^2$ = 0.584, F = 430.039, *p* = 0.000, service climate, and employee engagement were multiple mediation effects in the formation of customer behavioral intentions based on the SOR model. Figure 2 shows the regression coefficients for each path in the model. Therefore, H1–H7 of the hypotheses were all confirmed.

**Table 3.** Regression analysis of the mediation effects.

| Model | Variable | Model Parameters | | | Coefficient | | |
| --- | --- | --- | --- | --- | --- | --- | --- |
| | | R | $R^2$ | F | β | se | *p* |
| Model 1 | X→$M_1$ | 0.780 | 0.609 | 472.288 *** | 0.780 | 0.043 | 0.000 |
| Model 2 | X→$M_2$ | 0.843 | 0.710 | 369.392 *** | 0.233 | 0.055 | 0.000 |
| | $M_1$→$M_2$ | | | | 0.648 | 0.046 | 0.000 |
| Model 3 | X→$M_3$ | 0.917 | 0.841 | 529.002 *** | 0.125 | 0.047 | 0.001 |
| | $M_1$→$M_3$ | | | | 0.596 | 0.047 | 0.000 |
| | $M_2$→$M_3$ | | | | 0.249 | 0.047 | 0.000 |
| Model 4 | X→Y (control intermediaries) | 0.766 | 0.587 | 430.039 *** | 0.766 | 0.044 | 0.000 |

*** $p < 0.001$. X: SS, Y: BI, $M_1$: SC, $M_2$: CE, $M_3$: EE.

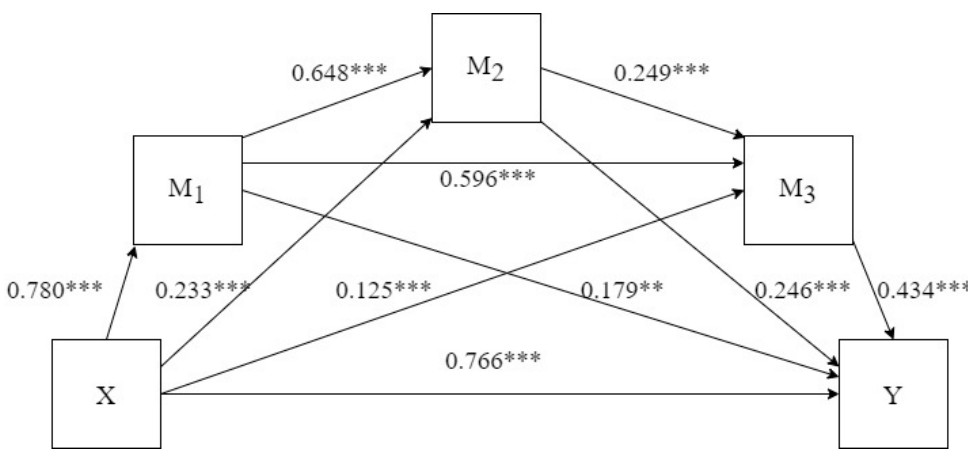

**Figure 2.** Validation model path of multiple mediated effects. Note: ** $p < 0.01$, *** $p < 0.001$.

### 4.2. Significance Test of Mediating Effect

The researchers used Bootstrap to test the significance of the mediation effect. First, 1000 Bootstrap samples were drawn from the original data (*n* = 305) using repeated random sampling. The conditioning values were used as the mean ±1 standard deviation. A 95% confidence interval for the mediation effect was estimated using the 2.5th percentile and 97.5th percentile. If the 95% confidence interval of these path coefficients did not include 0, a significant mediation effect would be indicated. As can be seen from Table 4, the 95% confidence interval for the total effect [0.599, 0.719] does not include 0, indicating that the model is reasonable. The 95% confidence interval of indirect effect Ind1 servicescape → service climate → behavioral intentions [0.029, 0.246] does not include 0, therefore service climate has a mediating role in the effect of servicescape on customer behavioral intentions; the confidence interval of indirect effect Ind2 servicescape → customer emotion → behavioral intentions [0.027, 0.098] does not include 0, indicating that customer emotion has a mediating role in the effect of servicescape on customer behavior; indirect effect Ind3 servicescape → employee engagement → behavioral intentions [0.015, 0.101] confidence

interval does not contain 0, indicating that employee engagement has a mediating role in the influence of servicescape on customer behavioral intentions; indirect effect Ind4 servicescape → service climate → customer emotion → behavioral intentions [0.074, 0.170] confidence interval does not contain 0, indicating that service climate and customer emotion have a mediating role in the influence of servicecape on customer behavioral intentions; the confidence interval of indirect effect Ind5 [0.143, 0.264] path does not contain 0, indicating that service climate and employee engagement have a mediating role in the influence of servicescape on customer behavioral intentions; the confidence interval of indirect effect Ind6 [0.012, 0.044] path does not contain 0, indicating that customer emotion and employee engagement have a mediating role in the influence of servicescape on customer behavioral intentions; the confidence interval of the indirect effect Ind7 [0.031, 0.085] path does not contain 0, indicating that the servicescape influences customer behavioral intentions through the mediating effects of service climate, customer emotion, and employee engagement; the direct effect servicescape → behavioral intentions [0.034, 0.224] confidence interval does not contain 0, indicating that all of the above mediated effects are partially mediated and that there are multiple mediated effects in this study. The behavior of customers coming to a resort hotel is influenced by the servicescape of the hotel. The servicescape directly affects the customer's willingness to patronize and positively enhances their behavioral intentions under the multiple mediating effects of the hotel's service climate, positive emotions, and perceived dedication of the staff.

**Table 4.** Bootstrap confidence interval effect parameters.

| Path | Effect | Bootstrap 95% Confidence Interval | |
| --- | --- | --- | --- |
| | | BootLLCI | BootULCI |
| Total | 0.658 | 0.599 | 0.719 |
| Direct | 0.129 | 0.034 | 0.224 |
| Ind1 | 0.140 | 0.029 | 0.246 |
| Ind2 | 0.057 | 0.027 | 0.098 |
| Ind3 | 0.054 | 0.015 | 0.101 |
| Ind4 | 0.125 | 0.074 | 0.170 |
| Ind5 | 0.202 | 0.143 | 0.264 |
| Ind6 | 0.025 | 0.012 | 0.044 |
| Ind7 | 0.055 | 0.031 | 0.085 |

Note: Direct: SS → BI; Ind1: SS → SC → BI; Ind2: SS → CE → BI; Ind3: SS → EE → BI; Ind4: SS → SC → CE → BI; Ind5: SS → SC → EE → BI; Ind6: SS → CE → EE → BI; Ind7: SS → SC → CE → EE → BI.

Because there were multiple concurrent mediators, in order to better understand the magnitude of the mediating role played by all the concurrent mediating variables together [66,67], the researchers conducted a two-by-two comparison (Table 5). The relative magnitude of the mediation effect was judged by comparing Bootstrap 95% confidence intervals between different path models. In Table 5, the standardized Bootstrap method test showed that the mediation effect Ind6 (servicescape → customer emotion → employee engagement → customer behavioral intentions) was significantly greater than the mediation effect Ind5 (servicescape → service climate → employee engagement → customer behavioral intentions) with an effect size of 0.177, and mediation effect Ind7 (servicescape → service climate → customer emotion → employee engagement → customer behavioral intentions) was greater than Ind5 with an effect size of 0.147, indicating that the effect of servicescape on customer behavioral intentions through the mediation of customer emotions and employee engagement was more significant in this model. This illustrated that, in the hot spring resorts, customer emotion and employee engagement were the main factors that determined whether they would visit again, and that a good service climate would stimulate employees to be more dedicated, and this effect improved the customers' service experience and made them more willing to share their experience or visit again.

**Table 5.** Comparison of Indirect Path Differences.

| Path | Effect | Bootstrap 95% Confidence Interval | |
|---|---|---|---|
| | | **BootLLCI** | **BootULCI** |
| Ind1 vs. Ind2 | 0.083 | −0.044 | 0.217 |
| Ind1 vs. Ind3 | 0.086 | −0.051 | 0.216 |
| Ind1 vs. Ind4 | 0.015 | −0.115 | 0.162 |
| Ind1 vs. Ind5 | −0.062 | −0.216 | 0.100 |
| Ind1 vs. Ind6 | 0.115 | −0.009 | 0.240 |
| Ind1 vs. Ind7 | 0.085 | −0.043 | 0.220 |
| Ind2 vs. Ind3 | 0.003 | −0.054 | 0.057 |
| Ind2 vs. Ind4 | −0.067 | −0.113 | −0.026 |
| Ind2 vs. Ind5 | −0.145 | −0.212 | −0.068 |
| Ind2 vs. Ind6 | 0.032 | 0.006 | 0.065 |
| Ind2 vs. Ind7 | 0.003 | −0.044 | 0.055 |
| Ind3 vs. Ind4 | −0.070 | −0.136 | 0.006 |
| Ind3 vs. Ind5 | −0.148 | −0.215 | −0.074 |
| Ind3 vs. Ind6 | 0.029 | −0.012 | 0.076 |
| Ind3 vs. Ind7 | 0.000 | −0.048 | 0.049 |
| Ind4 vs. Ind5 | −0.078 | −0.158 | 0.007 |
| Ind4 vs. Ind6 | 0.099 | 0.043 | 0.153 |
| Ind4 vs. Ind7 | 0.070 | 0.014 | 0.124 |
| Ind5 vs. Ind6 | 0.177 | 0.116 | 0.231 |
| Ind5 vs. Ind7 | 0.147 | 0.088 | 0.204 |
| Ind6 vs. Ind7 | −0.029 | −0.057 | −0.009 |

Note: Ind1: SS → SC → BI; Ind2: SS → CE → BI; Ind3: SS → EE → BI; Ind4: SS → SC → CE → BI; Ind5: SS → SC → EE → BI; Ind6: SS → CE → EE → BI; Ind7: SS → SC → CE → EE → BI.

## 5. Conclusions

This study used the SOR model, combining the hot spring resort environment, service, and customer emotions, and linking service environment, service climate, customer emotions, and customer behavioral intentions to construct the theoretical model. Through an analysis of 305 groups of data, we finally understood the influence of servicescape on customer behavioral intentions and the multiple mediation effect of customer emotions, service climate, and employee engagement. We hope that this approach can be used more widely in service-oriented industries.

### 5.1. Discussion

From the results of the correlation analysis, there were significant correlations between the variables examined (servicescape, customer emotions, customer behavioral intentions, service climate, and employee engagement), and these were all positive correlations, indicating that enhancing service landscape can directly or indirectly improve customer feelings in other areas as well as improve customer behavioral intentions. This finding is similar to previous studies, for example, in which Lin and Liang [58] concluded that the physical environment (e.g., environmental and design factors) was important for customer emotions and satisfaction with the business (e.g., fashion clothing stores) because they promoted pleasant emotional responses while enhancing customer perceptions. Lin [8] noted that servicescape significantly influenced customer emotions. Chang [36] found that the physical environment of hot spring resort hotels and the service of the hotel staff could make customers feel happy. In terms of the SOR model, positive customer perceptions of the physical environment of a business led to positive emotions or pleasurable feelings [68,69], which in turn positively influenced their satisfaction [70]. The positive perception of the hotel environment can also influence their subsequent behavioral intentions (e.g., revisiting, recommending to others, willingness to pay a higher price compared to competitors, etc.) [71,72]. The feelings of consumers were precisely attributable to stimulation by the servicescape in hotels [36,73,74], indicating that this should be an important focus for managers seeking to develop hotels in a sustainable way.

The model results show that servicescape has a positive impact on customer behavior through the multiple mediation effects of service climate, customer emotion, and employee engagement. Among them, the influence of servicescape on customers through the mediation effects of service climate and employee engagement is particularly significant. Customer emotions can influence employee engagement, and good customer mood can give employees a sense of accomplishment, which can in turn improve the quality of their service. Servicescape can promote customer behavioral intentions through a good service climate and high levels of employee engagement, which confirms the important influence of service climate and employee engagement on customer behavior. For this reason, service industries should pay greater attention to service climate and employee engagement, and this is consistent with the findings of previous studies. A better overall experience of a store can enhance the customer's perception of the physical environment and the experience of the service. Unlike other types of hotel, the core product of a hot spring resort is the hot spring bath, and the various auxiliary services and entertainment facilities associated with it are unique to a hot spring resort. Therefore, the role of servicescape in exploiting this valuable resource is all the more important [71]. Positive emotions can promote purchasing behavior. When consumers are in a good emotional state, they have a positive purchase intention. Therefore, positive emotions should be encouraged in consumers in order to promote the occurrence of consumption tendencies.

### 5.2. Research Limitations and Innovations

Our study is innovative in that previous studies of the service industry have tended to focus more on employee motivation and performance in order to increase sales or improve customer satisfaction. Only some parts of these studies have focused on the impact of servicescape [8]. Moreover, many of the studies have been aimed at the retail fields, such as stores, malls, and supermarkets. Very few researchers have investigated the service industry of hot spring hotels [8,75,76]. This study also considered the role of the external environment—servicescape—in directly and indirectly influencing customer emotions. The results of these studies demonstrated that the generation of human behavioral intentions is related to personality, biological and sociocultural experiences, goals, expectations, and internal and external factors [77].

As a result of time and cost constraints, some valuable data may have been overlooked in the selection of respondents for the questionnaire, for example, such as customers who did not complete the questionnaire. These might be customers who were dissatisfied with the service or were too demanding, and the service industry should also pay attention to the needs of such customers. In addition, the measurement of behavioral intentions was limited to the willingness expressed by the subjects rather than actual purchase or recommendation behavior; the frequency of repeat purchases or recommendations by validly measured customers would further enhance the validity of the study's findings. Finally, because of the differences in hotel types, each with its own characteristics, conclusions drawn from this specific research sample of a hot spring hotel have the potential to be biased. Future researchers will need to conduct data collection over a longer period of time and on a larger scale to construct a more stable mechanism of action model. Other variables, such as corporate image and perceived fairness, could be included in the research process to construct a comprehensive model of the mechanism of action, or a stratified regression study could be conducted within the service industry to investigate the influence of service scenarios and service climate on customer behavioral intentions more thoroughly.

### 5.3. Practical Implicaitons

According to our results, servicescape, service climate, customer emotion, and employee engagement can significantly influence customer behavioral intentions and have multiple mediation effects. Therefore, service industries should pay particular attention to the creation of their particular environments. Kaltcheva and Weitz [78] suggested that managers could provide different elements to stimulate emotional responses from customers,

such as music, color, and texture. Rosenbaum and Massiah [79] proposed that spa hotels could attract customers by designing different themes (e.g., tropical, oriental, modern minimalist) to provide a pleasant experience for customers. Furthermore, hotels should also increase recognition by maintaining their own style (e.g., romantic, nostalgic, exotic) by using color, music, and decoration [36]. Customers are stimulated by the servicescape of a range of amenities, which generates pleasure. This response is an important indication of a hotel's sustainability. A good environment not only influences customers, but also has a positive impact on employees. An excellent working environment can enhance employee creativity and further promote the development of the service industry. At the same time, operators should also focus on developing employee commitment and provide appropriate encouragement to employees [80]. When employees demonstrate excellent work and service performance, managers and employers should provide recognition and reward, as well as sufficient resources to enable employees to further develop their quality of service [8]. With regard to the customers, a quality service, and a relaxing and pleasant environment will increase their future behavioral intentions and willingness to pay a higher price in comparison to the competition. Hotel managers should be aware that customer experience is the result of a combination of the hotel environment and staff service.

### 5.4. Theoretical Implications

Based on the SOR theory, this study further enriches the research on customer behavioral intention and explores customer behavioral intention and influencing factors in the service industry represented by hotels. The process analysis shows that the research hypothesis is supported, and that the servicescape can predict customer behavioral intention; customer emotion plays a partial mediating effect in the influence of servicescape on customer behavioral intention; service climate and employee engagement play multiple mediating effects in the SOR model. The proposed theoretical model of "servicescape (stimulus)—customer emotion (body factor)—customer behavioral intention (response factor)" is valid, therefore the hotel can improve the customer's favorable feeling towards the hotel through decoration style and staff management to realize the sustainable development of the hotel.

This study will be of use to managers of hot spring resorts. The analysis reveals that hot spring resorts are able to achieve sustainable development not only from the servicescape, but also by creating a comfortable service climate, improving employee engagement and creating positive emotions and behavioral intentions among customers.

**Author Contributions:** Conceptualization, Z.G.; methodology, Z.G.; software, Z.G.; validation, Z.G.; formal analysis, Z.G.; investigation, Z.G.; resources, Z.G.; data curation, Z.G.; writing—original draft preparation, Z.G.; writing—review and editing, Y.-C.C. and Y.Y.; visualization, Z.G.; supervision, Y.-C.C.; project administration, Z.G.; funding acquisition, Z.G. All authors have read and agreed to the published version of the manuscript.

**Funding:** This research received no external funding.

**Institutional Review Board Statement:** Ethics approval is not required for this type of study. According to Article 3 of the Measures for Ethical Review of Biomedical Research Involving Human Beings of the People's Republic of China, ethical review is only required for biomedical research activities involving human beings, and our study did not use any physical, chemical, biological, or psychological methods to study human psychological behavior.

**Informed Consent Statement:** Informed consent was obtained from all subjects involved in the study.

**Data Availability Statement:** Data is not publicly available, though the data may be made available on request from the corresponding author.

**Conflicts of Interest:** The authors declare no conflict of interest.

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
