# Peer review of "Research on Customer Behavioral Intention of Hot Spring Resorts Based on SOR Model: The Multiple Mediation Effects of Service Climate and Employee Engagement"

_sustainability, doi:10.3390/su14148869_

Round 1

Reviewer 1 Report

Dear Authors,

The manuscript presents the results of interesting studies that were carried out in accordance with the prepared methodology. In my opinion, slight adjustments can be made.

Remarks:

1.     Figures and tables are properly prepared. I don't understand why there are yellow marks in Table 5.

2.     The presentation of the relationship between the theory and research results is insufficient. Worth extending.

3.     I am not sure if such a large number of research hypotheses is necessary. Perhaps this requires a justification.

4.     The references are appropriate. Perhaps it would be worth adding a few more recent items.

Reviewer 2 Report

Dear authors,

Hope my comments/suggestions can assist you in improving the manuscript. 

INTRODUCTION

·            The introduction part of this paper is well-written and the information provided appears to be significant. The research objectives are properly justified.

 LITERATURE REVIEW

·            Line 123-140. This paragraph focuses on the hypothesised relationship for H1. While, there is insufficient information evidence to support the relationship emotions and behavioral intention; servicescape and behavioral intention. Thus, H2 and H3 were developed with a minimal literature review, which is an area for improvement.

 METHODOLOGY

·            There was no discussion of common method variance solutions.

·            The control variables (e.g. age, gender, frequency of visits) were not included in the analysis? Please provide justification.

 RESULT

·            The findings are clearly presented and explained.

 DISCUSSION

·               Section 5.1. To avoid confusion with the section in section 4, please rename this section to “Discussion”.

 IMPLICATIONS

·               Section 5.3. This section should be renamed as “Practical implications”. Moreover, a section titled “Theoretical implications” is required to articulate the contribution of this study to theories, notably the SOR model.

Reviewer 3 Report

Although not perfect, it’s very well structured and reasoned. 

Author Response

Thank you for your very kind reply and comments!

Round 2

Reviewer 1 Report

Accept in present form